# Midterm Results of High-Dose-Rate Intraoperative Brachytherapy in the Treatment of Soft Tissue Sarcomas

**DOI:** 10.3390/cancers15102854

**Published:** 2023-05-21

**Authors:** Dietmar Dammerer, Johannes Neugebauer, Matthias Braito, Moritz Wagner, Markus Neubauer, Lukas Moser, Markus Süß, Michael Liebensteiner, David Putzer

**Affiliations:** 1Department of Orthopaedics and Traumatology, University Hospital Krems, 3500 Krems, Austria; 2Department of Orthopaedics and Traumatology, Medical University of Innsbruck, 6020 Innsbruck, Austria; 3Karl Landsteiner Private University for Health Sciences, 3500 Krems, Austria; 4Department of Orthopaedics and Traumatology, St. Johann in Tirol Hospital, 6380 St. Johann in Tirol, Austria; 5Department of Experimental Orthopaedics, Medical University of Innsbruck, 6020 Innsbruck, Austria

**Keywords:** high-dose brachytherapy, high-grade soft tissue sarcoma, intraoperative brachytherapy, soft tissue sarcoma of the extremities

## Abstract

**Simple Summary:**

Soft tissue sarcomas are rare malignant tumours that originate in the mesenchyme. Over 80 different entities have been identified to date. With an incidence of 4.7 and a median age of onset of 59 years, they account for about 1% of all malignant tumours in adults and 15% in children. The treatment regimen includes surgical resection, radiotherapy and, on a case-by-case basis, chemotherapy, which can achieve local control rates of about 90% for extremities and 50–80% for retroperitoneal sarcomas. In addition to external beam radiation, the use of intraoperative brachytherapy has for several decades offered the possibility of treating the exposed tumour bed directly with a single fraction of high-dose radiation, increasing its effectiveness while reducing toxicity to surrounding healthy tissue. This study investigated the rate of local recurrence in 35 individuals with soft tissue sarcomas of the extremity or retroperitoneum treated at the University Hospital for Orthopaedics Innsbruck from 2010–2016.

**Abstract:**

Introduction: According to the literature only sparse data are available on the use of high-dose-rate intraoperative brachytherapy (IOHDR-BT) as a boost to external-beam irradiation (EBRT) in combination with a wide resection in patients with high-grade soft tissue sarcomas (STS). Materials and Methods: Applying a retrospective study design, we investigated all patients who between 2010 and 2016 underwent marginal resection of a high-grade STS and intraoperative radiotherapy, followed by EBRT. We included only patients with a traceable follow-up time of at least two years. Of 89 patients, 35 met our inclusion criteria and showed an average follow-up of four years. Results: We found an overall 2-year local control rate of 94.3%. The local recurrence rate for R0 resections was 6%, whereas recurrences occurred in 13% of R1 resections and in 100% of R2 resections. One affected patient received only intraoperative radiotherapy. The recurrence rate by tumour entity was 36% for LPS, 11% for myxofibrosarcoma and 17% for undifferentiated pleomorphic sarcoma. Conclusion: The treatment regimen consisting of limb-preserving surgery, IORT and pre- or postoperative radiotherapy consistently shows excellent local control rates.

## 1. Introduction

Soft-tissue sarcomas (STS) comprise a heterogeneous group of tumours that represent 1% of tumours diagnosed in adults and account for over 20% of all pediatric solid malignant cancers [1]. According to the Surveillance, Epidemiology and End Results Program of the National Cancer Institute of North America, the incidence of STS is about 3.4 new cases per 100,000 inhabitants per year [2]. Men are affected more often than women, with a ratio of 1.4:1 [2]. In Europe, according to the RARECARE Project for rare cancers, the incidence is 4.7 per 100,000 population, with nearly 23,500 new diagnoses annually [3]. For Austria, an incidence of 2.4 per 100,000 inhabitants per year was calculated [4]. About one-third of sarcomas of the extremities and trunk are diagnosed superficially with an average size of 5 cm, while the other two-thirds are deep and already measure 9 cm at diagnosis [5]. Abdominal or retroperitoneal sarcomas are usually incidental findings in imaging procedures and then already have an average size of more than 10 cm [5]. Overall, 10% of all soft tissue sarcomas are metastasised at the time of diagnosis [5]. The lung is particularly affected due to the predominantly haematogenous spread [5].

While surgical resection of the primary tumor remains the mainstay of treatment for STS, function-sparing surgery supplemented with radiotherapy has been shown to be comparable to local control and survival rates compare with those for amputation [6]. Radiotherapy has traditionally been delivered with adjuvant external beam irradiation (EBRT). However, the use of EBRT is limited by radiotherapy-associated damage to the overlying skin and surrounding normal tissues [7,8] (complications related to radiation can be found in Table 1. The National Comprehensive Cancer Network guideline for STS recommends that perioperative irradiation be considered in the majority of patients after resection of a localized STS of the extremities and the superficial trunk [9]. According to the time scale needed to deliver the radiation dose (dose rate), BRT is nowadays available as an HDR (high-dose rate), an LDR (low-dose rate), and, recently, an ultra-low-dose rate. HDR can be administered twice a day for seven days as fractionated HDR, to overcome patients’ confinement and prolonged shielding [10,11]. Technically, administration is possible through flaps or seeds in the surgical bed or into the tumour mass (Figure 1) [10,11].

The therapeutic ratio adjuvant radiation may be further advanced by delivering additional irradiation to the surgical bed by means of brachytherapy (BT). The use of brachytherapy has already been shown to be useful in the treatment of high-grade sarcomas by reducing the risk of radiation-associated morbidity [12,13]. BT options comprise low-dose-rate techniques, fractionated high-dose-rate brachytherapy, or intraoperative high-dose-rate therapy (IOHDR) as a special form [14,15]. According to the American Brachytherapy Society consensus statement on intraoperative radiation therapy, this (IORT) represents an alternative to traditional EBRT that may shorten the course of therapy, reduce toxicities, and improve patient satisfaction while potentially lowering the cost of care. Although IOHDR has been used for many years in the treatment of sarcomas with close or positive resection margins or recurrent sarcomas, it has witnessed a resurgence in use more recently [16].

This monocentric retrospective study reports midterm results and aims to examine whether brachytherapy has additive value in the treatment of high-grade STS in the extremities.

## 2. Material and Methods 

The study protocol was approved by the Ethics Committee of the University of Innsbruck. Written informed consent was obtained from all patients before participation. All methods and measurements were performed in accordance with relevant guidelines and regulations. Inclusion criteria: The data analysis included patients who were treated with histo-pathological evidence of a high-grade sarcoma (grade II, III) at Innsbruck University Hospital between 2010 and 2016 and who had a documented clinical or radiological follow-up of at least two years. Patients who continued their follow-up treatment in the care of their home hospitals had to be excluded due to the limited data available. Exclusion criteria: Patients with grade I sarcomas were excluded because in our institution grade I sarcomas are only irradiated if they are located retroperitoneally, in risk areas that cannot be resolved surgically, if they are liposarcomas, or if there is already a recurrence. The local recurrence rate was evaluated, as well as time to local recurrence, metastasis rate, time to metastasis, overall survival rate, and survival time. Preoperative staging consisted of local magnetic resonance tomography (Figure 2 shows a leiomyosarcoma in the left thigh preoperatively, Figure 3 shows the same patient postoperatively), computed tomography of the whole trunk (thorax, abdomen, and pelvis), complete blood count, and serum chemistry analysis. Histologic diagnosis was assessed preoperatively by sonographically or CT-guided core needle biopsy. Inclusion criteria consisted of (a) surgical resection of the soft tissue sarcoma by wide surgical resection following the definitions by Enneking et al., [17] (b) combination with both IOHDR and EBRT and (c) follow-up time of at least two years. Patients who underwent amputation, intralesional resection with gross tumour remnants as well as patients with superficial low-grade sarcomas were excluded from the study. If neoadjuvant radiotherapy (percutaneous) was given at a dosage of 50 Gy (25 × 2 Gy) surgical rehabilitation could only take place after waiting out his side effects (Table 1). Adjuvant RT was around 60 Gy (30 × 2 Gy). This was scheduled after the removal of the stitches or when the wound was dry and healed. The sequence of treatment for the patients in the IOHD-BRT group was surgery and IORT followed by RT. The IOBT technique applied is a flap technique (Figure 1) [16]. 12–15 Gy are distributed over the flap surface using an iridium source (gamma emitter, iridium-192) with a very good dose profile. Within 1 cm there is a good dose fall-off and a relatively high dose can be administered once. After the wound has healed, a dose of 50 Gy has been administered percutaneously postoperatively (RT was started three to four weeks after tumour resection). Thus, intraoperative brachytherapy with 12–15 Gy plus 50 Gy postoperatively percutaneously results in a cumulative dose of 62–65 Gy. For percutaneous postoperative follow-up irradiation, brake radiation of photons from tungsten is used. External-beam radiation was delivered at a total dose of 50 Gy (given as 2 Gy per fraction) over five weeks encompassing the surgical bed with a margin of 3–5 cm. Patients suffering from high-grade sarcoma underwent postoperative doxorubicin-based adjuvant chemotherapy. Patients underwent regular follow-up evaluation at the authors’ institution every three months, consisting of clinical examination, computed tomography of the whole trunk (thorax, abdomen, and pelvis) and magnetic resonance imaging of the part of the body that had undergone surgery. During the above-mentioned investigated timeframe, we identified 89 patients who underwent surgical treatment at our department, combined with IOHDR and EBRT for soft tissue sarcoma of the extremities. Of these 89 patients, 35 showed a too-short follow-up period, which resulted from the fact that further therapy was administered at different hospitals, ten patients died during the observed period and nine patients showed a low-grade sarcoma in the resected tumour sample and therefore had to be excluded. Of the remaining 35 patients who met the inclusion criteria, 19 were male and 16 female, with an average follow-up of four years. Additional details are shown in the flow chart (Figure 4).

## 3. Statistical Analysis

Patient characteristics were descriptively analysed. Age at surgery is presented as mean with standard error (SE) and tumour size is presented as median with range. Sex, amputation, tumour location, tumour depth, tumour grade and whoops procedures are expressed as proportions. Local recurrence rate, metastasis rate, and overall survival rate are expressed as proportions and compared between treatment groups using Fisher’s exact test. Survival time, time to metastasis and time to local recurrence were estimated with the Kaplan–Meier method and differences between the treatment groups were compared with the log-rank test. Results were considered significant if *p* < 0.05. All statistical analyses were performed with SPSS (IBM SPSS Statistics for Windows, Inc, version 27.0; IBM Corp, Armonk, NY, USA).

## 4. Results

The age range of the patients in the collective was between 26 and 83 years at the time they underwent their tumour resection. Median age of the 19 men and 16 women was 67 years. The dose of IOHDR-BT was generally 15 Gy. In six (17%) cases, the surface dose was reduced by up to 10 Gy because critical structures (e.g., neurovascular bundles) were in close proximity to the applicator. For example, shown in Figure 2 a leiomyosarcoma in the left thigh starting with its origin in the femoral vein.

All but two tumours underwent additional EBRT with a total dose of 50 Gy, in 30 (86%) tumours postoperatively and in three (9%) preoperatively. In nine (26%) cases, chemotherapy was also included in the treatment plan. Two patients underwent neoadjuvant, five adjuvant and two others both pre- and postoperative chemotherapy. After primary surgical resection, nine (26%) patients with uncertain tumour histology were referred from other hospitals to Innsbruck Medical University Hospital for resection and intraoperative radiation. Specimen collection for histological typing was performed by excision biopsy in 13 (37%) tumours, sonographically guided biopsy in 21 (60%) and CT-guided needle biopsy once. Among the 35 high-grade tumours, 13 (37%) were Grade 2 and 22 (63%) were Grade 3 sarcomas according to the FNCLCC system (Fédération Nationale des Contre le Cancer) [18]. In 30 (86%) cases the primary tumour was treated, in four (11%) cases a local recurrence and in one case a lymph node metastasis of a liposarcoma was observed. In 17 (49%) operations, the resection margins (Table 2) were classified as R0, in 15 (43%) as R1 and in two (6%) as R2. In one case, no data were documented. Table 3 and Figure 5 show the different entities of observed sarcomas and their extremity distribution.

At a median of 3.9 years (range 0.5–4.8) after IOHDR-BT at Innsbruck Medical University Hospital, local recurrence occurred in six (17%) patients. Three women and three men were affected. The 2-year local control rate was 94.3% and was still 82.9% at the end of the observation period after a median of 3.8 (range 2–8) years. The local recurrence rate for R0 resections was 6%, whereby in one case an R1 operation with unknown tumour histology was performed at an external hospital and the patient was later referred to Innsbruck Medical University Hospital for R0 resection.

Furthermore, recurrences occurred in 13% of R1 resections and in 100% of R2 resections. One affected patient received only intraoperative radiotherapy. The other five additionally received postoperative EBRT and in one case neoadjuvant chemotherapy. Among the recurrent tumours were four Liposarcoma (LPS), one myxofibrosarcoma and one undifferentiated pleomorphic sarcoma. Thus, the recurrence rate by tumour entity was 36% for LPS, 11% for myxofibrosarcoma and 17% for undifferentiated pleomorphic sarcoma. In the extremities, the recurrence rate was 11%, and in the retroperitoneum 60%. Recurrence was seen in 8% of second-degree and 23% of third-degree sarcomas after treatment. Progression-free survival at 2 years was 80% (6% LR, 14% metastases) and at the end of the observation period 66% (17% LR and metastases each). Two (6%) patients died. One died at the age of 73 years (3.9 years postoperatively) due to recurrent sarcoma, and the other at the age of 58 years (2.6 years postoperatively) due to multilocal metastases. The two-year survival rate was thus 100% and overall survival over the full course of the study was 94%.

## 5. Discussion

The primary aim of treatment is to take a curative approach to achieve local tumour control (LC), i.e., local absence of radiolucent tumorcells, with the lowest achievable morbidity and limited functional impairment. According to the ESMO-EURACAN Clinical Practice Guidelines, surgical wide en bloc resection with R0 margins (Table 2 and Figure 6), i.e., removal of the tumour in one piece surrounded by healthy, tumour-free tissue, is the standard treatment to be aimed for in STS [20] Figure 6 shows a tumour bed after full tumor resection with R0 margins. Figure 7 shows the correlating tumor preparation. In Figure 1 one can see the inserted “Freiburger–Flap” (the radiation source).

With regard to the literature on IOHDR-BT in soft tissue sarcomas, several aspects need to be considered. Due to the rarity of the disease itself and because for several decades IOHDR-BT was available only in a few large centres, large prospective randomized trials are very rare. Most of the evidence is based on rather small retrospective analyses with very different observation periods and inhomogeneous patient cohorts. Often, no distinction is made between the different sarcoma entities, and the different localisations are also rarely taken into account. Often, recurrent or metastatic tumours are included. Furthermore, the quantity ratio of the different resection qualities, the different malignancy grades of the sarcomas as well as the use of further treatment options such as chemotherapy or regional hyperthermia varies from study to study, which further complicates comparison. The fact that soft tissue sarcomas are such a rare and at the same time so diverse group of diseases highlights the need for large, long-term, specific studies and makes them difficult to conduct. In previously published studies, the combination of surgery, IOHDR-BT and EBRT in patients with STS of the extremities consistently resulted in high 5-year local control rates of 82–97% and 40–89% for retroperitoneal STS [21,22,23,24,25,26,27,28,29,30,31,32]. Resection status and the proportion of recurrent disease treated were the most important determinants of local control [24,26,27,30] and, along with varying follow-up duration, were decisive for the wide variation in study results. In comparison, in this study at the Medical University of Innsbruck, the two-year rate of local control is 94.3% and over the entire observation period after a median of 3.8 (range 2–8) years it is 82.9%, combining STS of the extremities and retroperitoneum. When limb and superficial trunk sarcomas are considered separately, the local control rate over the entire study period is 90%. There is only one study, namely by Roeder et al., that comes to a much better result of 97%, with only 12% of R1 resections [33]. In contrast, 43% of R1 resections and 6% of R2 resections are included in this work. In other comparable studies, the proportion of R+ resections (R1 + R2) is 12–39% [33,34,35,36,37,38,39]. Retroperitoneal sarcomas have local control of 40% in a very small cohort of only five patients, which is still much better than the 20% achieved with postoperative radiation alone [21]. The result of this study is also remarkable in view of the fact that 26% of the affected patients were referred to Innsbruck Medical University Hospital only for resection after an external R1 resection, as this represents a negative predictive value for the further course of the disease. The incidence of wound healing disorders (surgical debridement 6%, surgical revision and split skin coverage 11% each) and radiodermatitis (57%) increased compared to other studies. In many studies, however, the type of complication is not described in detail, but only as a skin complication (42%) or wound complication (15–17%), which in turn makes comparability more difficult [36,38,39,40]. The follow-up period of at least two years was chosen because 70–80% of local recurrences are diagnosed in the first two to three years after completion of therapy [41,42]. However, because these tumours can recur even after a long progression-free period, the longest possible follow-up should be aimed for. In addition to grading, the status of the resection margin proved to be the most important prognostic factor for the occurrence of local recurrences, as in many other studies [39,43,44,45]. In Figure 6 we present a tumour bed after tumor resection. In Figure 7 we want to show how large the block to be resected has to be in order to remain tumour-free margins. In some cases, plastic coverage may be necessary here. The importance of the resection margins becomes clear in comparison to Table 4. However, often no distinction is made between the three malignancy grades, but only between low-grade and high-grade sarcomas. The situation is different with tumour size, which is considered a negative predictive value for local control from 5 cm, which showed no influence here [46,47,48,49]. That assumption that IOHDR-BT has a positive influence on the local control rate seems to be assured [41,42,46], but no consensus has yet been reached on the question of whether it can also influence overall survival. There are studies that show improved survival, especially in retroperitoneal and colorectal sarcomas, as well as those that do not confirm this assumption [21,31]. For example, in retroperitoneal and colorectal sarcomas, the combination of preoperative RT, R0 resection and IOHDR-BT has shown not only improved local control but also a positive impact on 5-year overall survival as compared with the non-IORT regimens in some non-randomised comparative studies [21,23]. STS of any location was also associated with a 40% reduced relative risk of death or recurrence in a review of 251 patients from Germany, 92 of whom received IORT [49]. In the synopsis of existing literature and our own results, the combination of limb-sparing surgery, IORT and pre- or postoperative radiotherapy consistently produces excellent local control rates in extremity STS that are at least comparable to approaches using EBRT alone and generally include patient cohorts with higher proportions of less severely affected patients with better prognosis (Table 5).

In addition, approaches involving IORT result in very high limb preservation rates and good functional outcomes, probably related to the lower high-dose volume. In retroperitoneal STS, the combination of preoperative EBRT, surgery and IORT consistently achieves high local control rates, which appear to be superior to surgery alone or surgery with EBRT, at least in terms of local control and even overall survival in some reports. Furthermore, preoperative EBRT in combination with IORT appears to be superior to combination with postoperative EBRT in terms of local control and toxicity. No major differences in wound healing disorders or postoperative complication rates are observed with IORT as compared to approaches without IORT. Neuropathy of the major nerves remains a limiting factor requiring dose restrictions or exclusion from the target area. Gastrointestinal structures and ureters should be excluded from the IORT area if possible and the IORT volume should be limited to the minimum available. Nevertheless, IORT in combination with external beam irradiation is an ideal method for saturation, allowing the administration of very high doses with low toxicity. In order to be able to more individually predict the probability of relapse for the different sarcomas and their different characteristics such as severity and resection status, as well as to confirm the assumption of improved overall survival by IORT, further preferably large prospective randomized studies are necessary. Our data are also very limited. Despite the positive data supporting IORT, brachytherapy is not a standard method used for the complementary treatment of soft-tissue sarcomas [10,51]. The biggest problem seems to be its limited accessibility and technical limitations resulting from the fact that brachytherapy facilities are rarely located near surgical or oncologic orthopaedic departments offering the possibility of a shared surgical path [10,51,52]. The recent introduction of portable linear accelerators, delivering low-energy (50 kV) photons, could be an option for solving this limitation [10,53,54].

## 6. Conclusions

The treatment regimen consisting of limb-preserving surgery, IORT and pre- or postoperative radiotherapy consistently shows good local control rates. Thus, the local control rate is 94.3% and 82.9%, respectively. Considered separately, the local control rate over the entire course of the study is 90% for extremity sarcomas. We find that even with good local control, the risk of metastasis in high-grade STS could not be reduced by this therapeutic strategy. The surgical resection of the tumour with a safety margin of healthy tissue is the fundamental method in the treatment of soft-tissue sarcomas. However, in large soft-tissue sarcomas or sarcomas that cannot be completely surgically resected (for example, if they are retroperitoneal or in areas at risk), therapy should include a combination of surgical intervention and radiotherapy. In our opinion, brachytherapy is preferable, when possible. Adjuvant brachytherapy increases the local control rate to up to 78%, is well tolerated, and rarely causes complications. Treatment should be delivered in specialist centres with multidisciplinary resources [10,51,55].

## Figures and Tables

**Figure 1 cancers-15-02854-f001:**
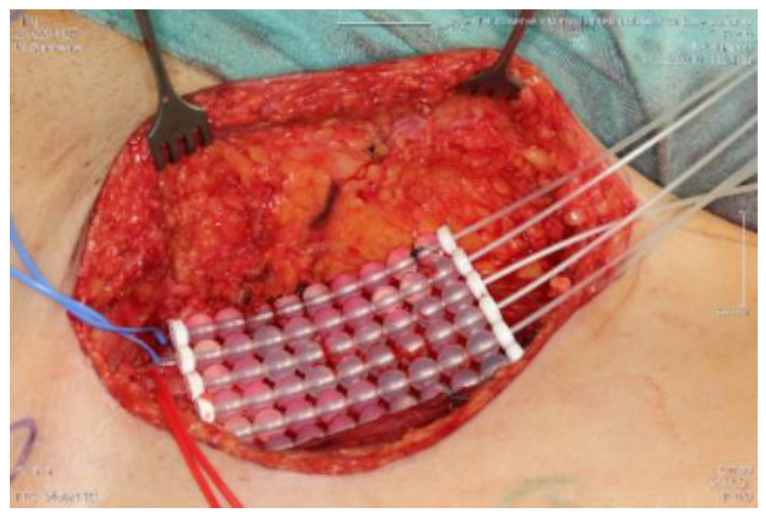
Intraoperative irradiation of the tumour bed; “Freiburg Flap” applicator.

**Figure 2 cancers-15-02854-f002:**
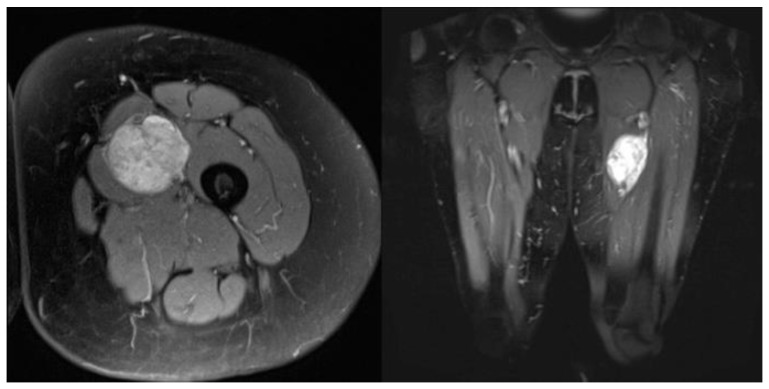
MRI axial (left) and coronal (right) of a leiomyosarcoma in the left thigh starting from the femo-ral vein.

**Figure 3 cancers-15-02854-f003:**
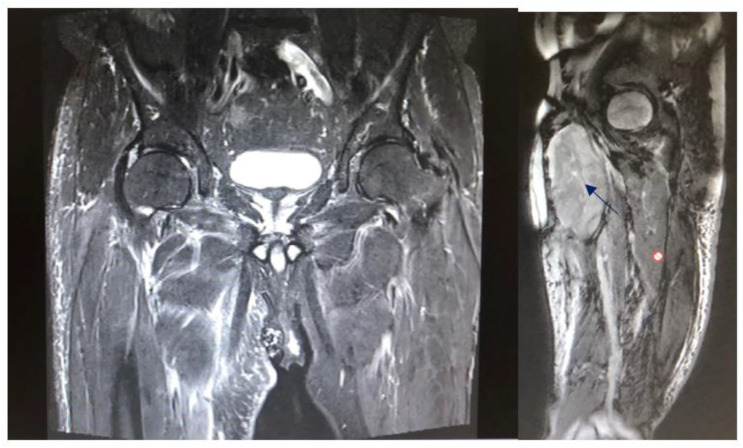
Patient after tumor resection: (left: 2 months postoperatively tumor-free; right: initial after sur-gery with a haematoma (blue arrow) and scar tissue (red dot).

**Figure 4 cancers-15-02854-f004:**
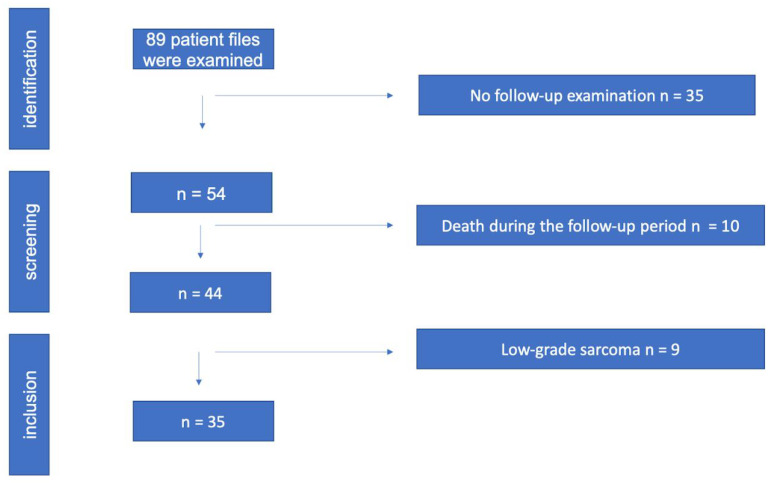
Inclusion and exclusion criteria.

**Figure 5 cancers-15-02854-f005:**
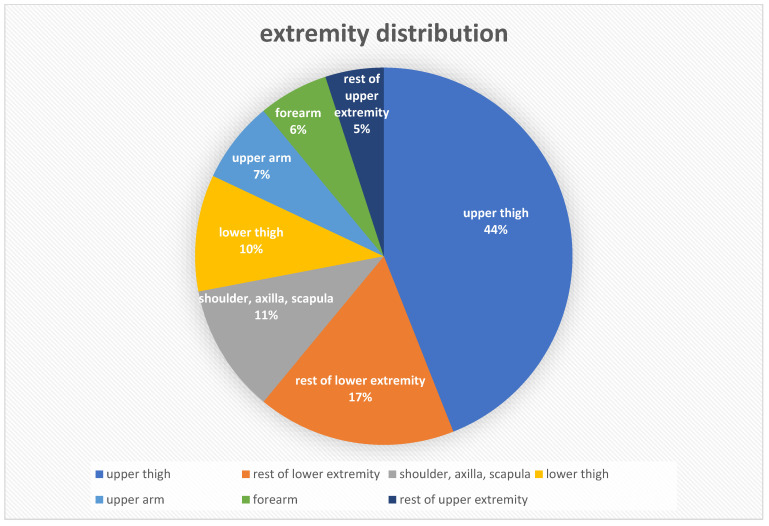
Extremity distribution in the current study.

**Figure 6 cancers-15-02854-f006:**
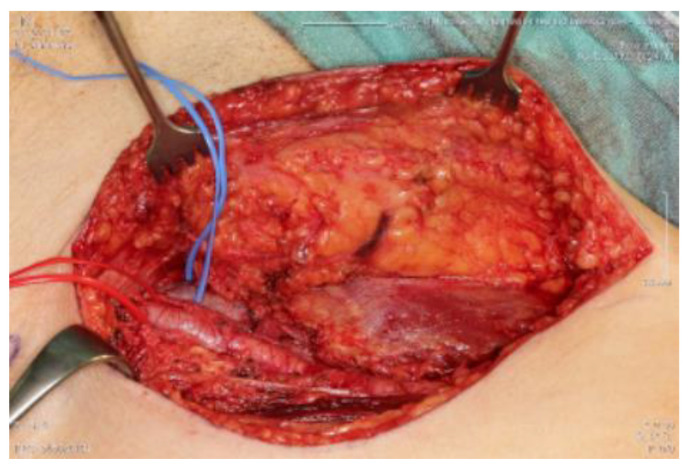
Tumour bed following sarcoma resection.

**Figure 7 cancers-15-02854-f007:**
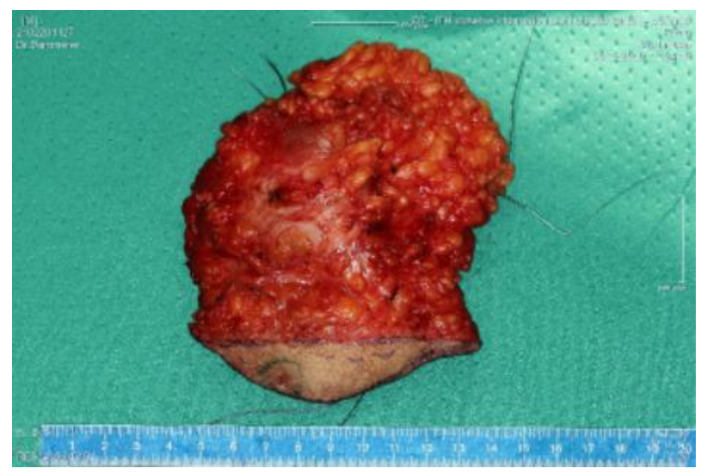
Resected tumour.

**Table 1 cancers-15-02854-t001:** Radiotherapy-associated complications (postoperative).

Complication	Number (%)
Radiodermatitis	20 (57)
Stage 1	15 (43)
Stage 2	4 (11)
Stage 3	1 (3)
Surgical Debridement	2 (6)
Surgical Revision	4 (11)
Splitskin Coverage	4 (11)

**Table 2 cancers-15-02854-t002:** Status of resection margin AJCC [19].

Status	Description
R0	No residual Tumor
R1	Residual tumour microscopically detectable/resection along the pseudocapsule
R2	Residual tumour macroscopically visible
Curative tumour resection = R0 in the absence or after resection of metastases	

**Table 3 cancers-15-02854-t003:** Different types of Sarcoma included.

Types of Sarcoma	*n* = 35 (%)
Liposarcoma	11 (31)
Pleomorphic undifferentiated sarcoma	9 (26)
Myxofibrosarcoma	6 (17)
extraskeletal myxoid chondrosarcoma	2 (6)
Myofibrosarcoma	2 (6)
Synovial sarcoma	2 (6)
Rhabdomyosarcoma (pleomorphic)	1 (3)
Leiomyosarcoma	1 (3)
Malignant peripheral nerve sheath tumour	1 (3)

**Table 4 cancers-15-02854-t004:** Resection status and entity.

Resection Status	Number of LR (%)	No LR (%)	Total
R0	1 (6)	16 (94)	17
R1	2 (13)	13 (87)	15
R2	2 (100)	-	2
Lost to follow-up	1 (100)	-	1
Types of Sarcoma			
Liposarcoma	4 (36)	7 (64)	11
UPS	1 (11)	8 (89)	9
Myxofibrosarcoma	1 (17)	5 (83)	6
Other entities	-	9 (100)	9
Localization			
Extremities	3 (11)	25 (89)	28
Retroperitoneum	3 (60)	2 (40)	5
Superficial trunk	-	2 (100)	2
Size			
<5 cm	2 (18)	9 (82)	11
>5 cm	4 (17)	20 (83)	24
Grading			
2	1 (8)	12 (92)	13
3	5 (23)	17 (77)	22
Staging			
T1 N0 M0	2 (18)	9 (82)	11
T2 N0 M0	4 (27)	11 (73)	15
Other class	-	9 (100)	9
Additional therapy			
Adjuvant RT	5 (15)	28 (85)	33
Chemotherapy	1 (11)	8 (89)	9

**Table 5 cancers-15-02854-t005:** TNM classification for STS [50].

TX	Primary tumours can not be assessed
T0	No evidence of primary tumour
T1	≤5 cm in greatest spread
T2	5–10 cm
T3	10–15 cm
T4	>15 cm
N0	No regional lymph node metastases
N1	regional lymph node metastases
M0	No metastases
M1	Metastases

## Data Availability

The data are not publicly available due to privacy.

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
