# Peer review of "Midterm Results of High-Dose-Rate Intraoperative Brachytherapy in the Treatment of Soft Tissue Sarcomas"

_cancers, 2023, doi:10.3390/cancers15102854_

Round 1

Reviewer 1 Report (Previous Reviewer 2)

The authors have made revisions to this manuscript and have improved it overall. 

For the introduction, would not delete the entire section that discussed brachytherapy and IOHDR background and rationale for its use. The authors had previously noted that IOHDR may allow for dose reduction of EBRT but never demonstrated any dose reduction in their cohort. This still should be presented as a rationale for its use but they just need to note this in their methods and discussion.  Current strategies in improving radiation therapy as a whole in STS involve efforts to decrease the length of therapy as well as toxicity. This is the framework that the authors should continue to place their series/experience. 

As of now - there has been no demonstrated improvement in outcomes with a "boost" to preop EBRT strategies. As stated in past review this should be noted and again would frame IOHDR as a potential method for dose reduction? 

This statement by the authors at the end of the Introduction should be reworded : "This monocentric retrospective study reports midterm results and aims to examine whether brachy-77 therapy has additive value in the treatment of high-grade STS in the extremities." -- this study is not designed (as no control or comparator group) to answer this question. This study is reporting of the use of brachytherapy based on single center experience on use high-dose IOHDR-BT as a boost to EBRT and can only be used to highlight local control outcomes, toxicity/complications of treatment, and survival in this series.  It can't be used to say that this is superior to other strategies. 

Author Response

Reviewer 2 Report (New Reviewer)

The manuscript by Dammerer et al retrospectively analyzes the patients diagnosed with high-grade soft tissue sarcomas of the extremity or retroperitoneum who have been treated at the authors' institution from 2010 to 2016 with intraoperative high dose-rate (HDR) brachytherapy as a boost to external-beam radiotherapy.  

It is a very interesting study showing promising results on the local control rates of brachytherapy combined with limb-preserving surgery and pre- or postoperative radiotherapy, further confirming the role of brachytherapy in the treatment of sarcomas. Thus, it is acceptable for publication with minor revision.

Minor comments:
Line 32: remove "thus"
Line 55-62: move this section to the discussion
Line 56: "i.e." instead of "means"
Line 73-76: move to line 65 after "amputation".
Line 84: a histopathological evidence
Line 354: therapeutic strategy

Author Response

This manuscript is a resubmission of an earlier submission. The following is a list of the peer review reports and author responses from that submission.

Round 1

Reviewer 1 Report

This retrospective study by Dammerer et al. examines the response of high-grade soft tissue sarcomas to intraoperative radiotherapy and subsequent external-beam irradiation after tumor resection. While similar studies appear to have been published previously, this effort is distinguished, at least to some degree, by breaking down the response to this radiotherapy on the basis of resection margins, the type of sarcoma, the location, size, tumor grade and stage, and whether or not adjuvant chemotherapy was administered. This work should be of particular interest to radiation oncologists and those caring for sarcoma patients. Nevertheless, there are some issues related to presentation that should be addressed prior to publication.

1. Most importantly, the last 3 lines of p. 5 (beginning "In existing studies, the combination...") makes it clear that that multiple other studies have combined surgery, IOHDR-BT and EBRT in patients with soft tissue sarcomas.  What is unclear here, in the Discussion, as well as in other places, is what the current study will add to the literature and why it will have value to clinicians and researchers. Related to this, it is often unclear whether statements refer to the current study alone or the current study together with previous studies (an example would be "In summary, the combination of limb-sparing surgery, IORT and pre- or postoperative radiotherapy consistently produces excellent local control rates..." in lines 15-17 from the bottom of p. 6).

2. The text does not refer to Figures 1 & 3-4 or Table 3.  This should either be corrected or these figures/tables removed. 

3. There are some minor formatting issues. Among these, numbers indicating author affiliations after their names should use subscripts and a corresponding author should be identified. Additionally, references in the last line of page 1 "(6) (7)" do not appear to be formatted correctly.

4. There are multiple places where the terminology utilized is either awkward, needs to be more fully defined (at least the first time it is utilized), or made clearer for a more general reading audience. This includes "marginal resection" in the 3rd line of the Abstract, "...are diagnosed in a local stage" in the 3rd line of the Introduction, "...radiotherapy has been shown to be comparable to local control" (clarification on what constitutes local control appears to be needed) in the 7th-8th lines of the Introduction, "...compare with those for ablative surgery" (better definition of ablative surgery appears to be needed) in the 8th line of the Introduction, "ablative surgery" (needs definition there or earlier) in the 15th line of the Materials and Methods, "whoops procedures" (needs better definition for a more general audience) in the 3rd line of the Materials and Methods on p. 3, "local control rate" in the 5th line of p. 4, "second-degree" and "third-degree" sarcomas (it is unclear what this refers to) 6 lines before the start of the Discussion on p. 5, "external R1 resection" (how does an "external R1 resection" differ from a "R1 resection"?) in the 15th line of p. 6.

5. There are additional instances of terms or acronyms not being defined. This includes "FNCLCC" not being defined before it is first utilized in the 4th to last line of p. 3, utilization of the terms "R0"/"R1"/"R2" to describe surgical margins in both the Abstract and the last line of p. 3 without indicating what they refer to, the utilization of "LR"/"UPS"/"T1 N0 M0"/"T2 N0 M0" in Table 3, and "LPS" in multiple places.

6. There are some apparent typos: should "the" be "they" in the 2nd line of the Results section; should "comparability" be "comparisons" in the 20th line of p. 6; should "grading" be "tumor grade" in the 24th line of p. 6?

7. The statement "That assumption that IOHDR-BT has a positive influence on the local control rate seems to be assured..." is awkward and unclear.

8. "...40% reduced relative risk..." in the 18th to last line on p. 6 is in comparison to what? 

9. It would be useful if the title to Figure 2 indicated the study (current and/or others) this refers to.

Reviewer 2 Report

Summary:

This manuscript is a retrospective case series of 35 patients (2010 – 2016 undergoing intraoperative brachytherapy followed by EBRT after marginal resection of STS. The authors sough to determine local control results with the use of intraop brachytherapy as an “additive booster therapy” in the treatment of high-grade STS of the extremity.

In this series, the 2-year local control rate was 94.3%, local recurrence rate for R0 resections was 6% versus 13% for R1.

Major Issues:

In the abstract, the authors note intraop brachytherapy after marginal resection and then later refer to margins with “R” classification. The authors need to clarify their margin description and be consistent throughout the paper.

In the introduction the authors reference the NCCN guidelines recommendations for treatment of extremity STS to undergo postoperative RT in majority of patients after resection. This is somewhat inaccurate as the recommendation is for either neoadjuvant or adjuvant, but as per the RT specific discussion the panel recommendation is for neoadjuvant RT: “Based on the pros and cons of neoadjuvant versus adjuvant radiation, the panel has expressed a general preference for neoadjuvant RT, because adjuvant radiation fields are typically larger than neoadjuvant radiation fields.” NCCN STS 2022.

In the introduction the authors note potential benefit of strategy of intraop brachytherapy being reduction in EBRT (dose and treatment length).  The authors should further comment on this strategy. Doesn’t appear that in this series this was done and that EBRT dosing was not reduced.

While local control data is important to review local control modality - there is a lack of presentation of complications of treatment. The authors need to add this to both intro, results, and discussion. 

In the discussion the authors’ first paragraph is too long and requires revision. The first paragraph should summarize key findings of the current study and importance of this paper. Then subsequent paragraphs can be used to compare and place in context of rest of literature.  

The key issues the authors need to further address is how the addition of IOHDR-BT to established EBRT/IMRT regimens adds further local control without adding to complications/morbidity of treatment.

Previous studies have demonstrated superiority of IMRT over brachytherapy in treatment of STS (Alektiar et al Cancer 2011). The authors should further compare previous studies/data on outcomes between these RT modalities.  

Thus far, there has been no compelling evidence for the addition of a “boost” to RT treatment. Previous papers have demonstrated no significant benefit with the addition of a boost in the setting of preoperative RT and resection with positive margins (Pan E J Surg Oncol 2014, Al Yami A Int J Radiat Oncol Biol Phys 2010, Haas RL and Gronchi A Radiotherapy and Oncology 2021) additionally good local control outcomes have remained in the setting of R1 margins along critical structures following preoperative radiation therapy (O’Donnell PW Cancer 2014).

The authors do not have a limitations paragraph in the discussion. This should be included with discussion of how there is no comparative group for comparison that received EBRT only and did not undergo intraop brachytherapy.

The last paragraph of the discussion focuses on IORT – even in retroperitoneal STS. There are no references cited in the entire paragraph despite comments to established outcomes. If the authors are going to reference retroperitoneal STS and RT – could reference the STRASS trial.

Minor Issues:

In the results section, histologic subtypes include liposarcoma, which subtypes were included with this?

The authors need to further clarify patients receiving neoadjuvant or adjuvant chemotherapy. In the methods the authors state: “patients suffering from high-grade sarcoma underwent postoperative doxorubicin-based adjuvant chemotherapy.” This requires further detail as to chemotherapy use and agents (doxorubicin and ifosfamide vs doxorubicin alone).

Reviewer 3 Report

Summary
I name the whole manuscript in a single word - chaos. It’s very hard to understand the treatment sequence and results. I feel that various information is randomly described within the text. Moreover, the conclusions do not correspond with the aim and methods. The analyzed cohort is small, retrospective, and very heterogeneous.
I kindly suggest that the paper in its present form should not be approved for publication, and I recommend rejection.

Introduction
- Provide a reference for “Approximately half of such patients are diagnosed in a local stage”
- What does mean “ablative surgery”? Do you mean definitive/radical surgery?
- “The National Comprehensive Cancer Network guideline for STS recommends that postoperative irradiation be considered in the majority of the patients after resection of a localized STS of the extremities and the superficial trunk” - it is not true. NCCN guidelines and ASTRO guidelines recommend PERIoperative radiotherapy, not postoperative. Moreover, preoperative radiotherapy is a preferred approach (see: https://doi.org/10.1016/j.prro.2021.04.005 and https://doi.org/10.3390/cancers12082061).

Materials and methods
- Why did you exclude patients with low-grade sarcomas? These tumors are more radioresistant than G2 and G3… (Table 1).
- As I understood you included patients treated outside your center (recurrent after whoops surgery)? Please provide CLEAR inclusion and exclusion criteria.
- “Neoadjuvant radiotherapy was given at a dosage of 50 Gy (25 × 2 Gy) within a time limit of four to six weeks after surgery.” --> I am sorry but I don’t understand this. Did you give NEOadjuvant RT BEFORE surgery?
- The whole paragraph from “Neoadjuvant radiotherapy…” up to “Patients suffering from high-grade sarcoma” is very confusing. Please rewrite it and add a figure with the CLEAR diagram presenting the treatment sequence (all methods - EBRT, IORT, chemo, etc.).

- “Adjuvant RT was around 60 Gy (30 × 2 Gy).” - it is not very precise and could be significant for this study. Please describe it more accurately (exact doses, target volumes, etc.).
- What about EBRT techniques? 2D? 3D? IMRT/VMAT? Protons?

Results
- “All but two tumors underwent additional EBRT with a total dose of 50 Gy, in 30 (86%) tumors postoperatively and in three (9%) preoperatively.” - but above you wrote that “the sequence of treatment for the patients in the IOHD-BRT group was surgery and IORT followed by RT.” (and in the abstract “a high-grade STS and intraoperative radiotherapy, followed by EBRT”). So… what was the treatment sequence? There you stated that three patients received preoperative RT. I’m sorry but I totally don’t understand it.
- “After primary surgical resection, nine (26%) patients with uncertain tumor histology were referred from other hospitals to Innsbruck Medical University Hospital for resection and intraoperative radiation.” - after “whoops” (R1) surgery or with recurrent tumors? Again, please add a CLEAR figure that presents and summarizes the results.
- Can you say that the event after R2 surgery is a recurrence? Rather progression (you have a macroscopic tumor!).

Discussion
- I am not assessing discussion until the rest of the manuscript will significantly improve.

Conclusions
Conclusions are overstated. The aim of your study was to assess if IORT provides any extra value to EBRT in terms of local control. Your methodology did not allow you to answer this question (single-center, single-arm, retrospective). Compared to historical cohorts/trials with EBRT-alone results look similar. Thus, does IORT provide any additional value? No.
Moreover, I don’t understand the last sentence: “We find that even with
very good local control, the risk of metastasis in high-grade STS could not be reduced by this therapy scheme”.

Tables
Table 1 - it is rather a figure than a table. Why have so many patients had no follow-up examinations? It’s very surprising.
Table 2
- “Sarcomentity” --> what does it mean? It’s a new word because even Google could not find it.
- Myofibrosarcoma - please use the updated classification.
- Rhabdomyosarcoma - what subtype? It’s a significant difference between pleomorphic and embryonal RMS.

Figures
Figure 1 - provide any follow-up MRIs after treatment to show surgical bed.
Figure 2 - poor quality (some of the text is not visible) and, honestly, unnecessary.

Round 2

Reviewer 2 Report

The authors have made attempts to address reviewer comments; however, there are some issues that still remain. 

I would recommend the authors exclude retroperitoneal disease and only evaluate extremity and trunk.

The authors in Table 3 use size cutoff of < or > 5 cm, in line with older AJCC staging, but in Table 6 TNM system they use T1-T4 as used in the AJCC 8th edition. The authors should comment on this discrepancy and why the different size cutoffs were used for different analysis. 

Paragraph 1 of the discussion is still too extensive and should be revised to help organize the discussion section. 

It appears that only 3 (9%) of patients recieved preop RT - but yet the conclusion paragraph highlights local control outcomes between preop and postop RT. 

The authors' conclusion paragraph merely restates local control rates - comparing preop at 94.3% vs postop at 82.9%. This should be addressed in discussion section with comments on why different local control results as well as discussion regarding sequencing of EBRT and role of IORT within that. The conclusion paragraph needs to be rewritten in response to results. 

The authors had mentioned benefit of dose reduction of EBRT required in setting of IORT with brachytherapy. However, it appears that adjuvant (postop) RT was still given at 60Gy. Is this a dose reduction compared to standard adjuvant RT given at this institution for those patients that didn't received IORT brachytherapy?

Reviewer 3 Report

Summary
This is a revised version of the previously assessed manuscript.
After careful reassessment, I maintain my previous opinion. I kindly suggest that the paper in its present form should not be approved for publication, and I recommend rejection.

I thank the authors for performed corrections. However, the quality and scientific validity of the manuscript is not satisfactory to me.

Remaining issues (based on the table with responses provided by the authors):

1) OLD COMMENT: "Adjuvant RT was around 60 Gy (30 × 2 Gy)." - it is not very precise and could be significant for this study. Please describe it more accurately (exact doses, target volumes, etc.).
What about EBRT techniques? 2D? 3D? IMRT/VMAT? Protons?

NEW COMMENT:
You did not provide the data I asked for (statistics of used techniques, contouring guidelines, margins, etc.), and only added some terms to the glossary.

2) OLD COMMENTS: As I understood you included patients treated outside your center (recurrent after whoops surgery)? Please provide CLEAR inclusion and exclusion criteria. AND Why did you exclude patients with low-grade sarcomas? These tumors are more radioresistant than G2 and G3... (Table 1).

NEW COMMENT: I am asking again for the explicit inclusion and exclusion criteria.

3) OLD COMMENT: "Neoadjuvant radiotherapy was given at a dosage of 50 Gy (25 × 2 Gy) within a time limit of four to six weeks after surgery." --> I am sorry but I don't understand this. You gave NEOadjuvant RT BEFORE sugery?

AUTHORS' RESPONSE: At this point we are unsure if we
understand the comment correctly and
ask for a more insightful explanation.
Yes, we gave the RT neoadjuvant
before the surgical procedure. We do
not understand the Fact, because the
dose is distributed to radiant

NEW COMMENT: Your explanation seems to be unfinished. Your sentence suggests that you gave neoadjuvant radiotherapy 4-6 weeks after surgery. It's not correct.

4) OLD COMMENT: The whole paragraph from "Neoadjuvant radiotherapy..." up to the "Patients suffering from high-grade sarcoma" is very confusing. Please rewrite it and add figure with the CLEAR diagram presenting treatment sequence (all methods - EBRT, IORT, chemo, etc.).

NEW COMMENT: You did not comment on nor provided the requested clarification.

5) OLD COMMENT: Conclusions are overstated. The aim of your study was to assess if IORT provides any extra value to EBRT in the term of local control. Your methodology did not allow to answer this question (single center, single arm, retrospective). Compared to historical cohorts/trials with EBRT-alone results look similar. Thus, does IORT provide any additional value? No.

NEW COMMENT: Thank you for the change to "good" but the conclusions do not match the aim of your study. You asked whether "brachytherapy has additive value in the treatment" (it was the aim of the study), not whether it provides good local control.

6) OLD COMMENT: Moreover, I don't understand the last sentence: "We find that even with
very good local control, the risk of metastasis in high-grade STS could not be reduced by this therapy scheme".

NEW COMMENT: You have not discussed this point. It was not your study's question (endpoint, aim, etc.). It's also not the finding from this study.

7) OLD COMMENTS:
Tables
Table 1 - it is rather figure than table. Why so many patients had no follow-up examinations? It's very suprising.
Table 2
"Sarcomentity" --> what does it mean? It's a new word because even Google could not find it.
Myofibrosarcoma - please use the updated classification.
Rhabdomyosarcoma - what subtype? It's a significant difference between pleomorphic and embryonal RMS.

NEW COMMENT:
You have not considered changing the chart into the figure, nor commented on it.
"Myofibrosarcoma" is in Table 1  (and is separated from myxofibrosarcoma). According to the WHO Classification of Tumours of Soft Tissue and Bone, such a diagnosis does not exist.

8) OLD COMMENT: Figure 1 - provide any follow-up MRIs after treatment to show surgical bed

AUTHORS' RESPONSE: Thank you very much, of course we
can submit postoperative pictures later,
which picture in particular would be of
interest ?

NEW COMMENT: at 6, 12, and 24 months after the completion of treatment, please add them as supplementary files to the article, thank you.

My additional remarks:
You stated that "However, we would like to point out that our results do not claim to be a comparison, we only describe therapies that have already been performed. " Thus, in my opinion, it is not possible to evaluate whether brachytherapy has additive value in the treatment of high-grade STS in the extremities. How would you like to measure the added value of brachytherapy to EBRT? As a result, your methods and sample did not allow you to answer your question.

The quality of figure 2 is still poor, it's not about resolution but the overall project --> "rest of upper extremity" is not visible (white color on a bright background).